Temporal whole field sawtooth flicker without a spatial component elicits a myopic shift following optical defocus irrespective of waveform direction in chicks

Murphy Melanie J. m.murphy@latrobe.edu.au 1
Riddell Nina 1
Crewther David P. 2
Simpson David 3
Crewther Sheila G. 1
1 School of Psychology & Public Health, La Trobe University , Melbourne , Victoria , Australia
2 Centre for Human Psychopharmacology, Swinburne University of Technology , Melbourne , Victoria , Australia
3 Brain Sciences Institute, Swinburne University of Technology , Melbourne , Victoria , Australia
Macknik Stephen
Electronic publication date: 2019 Jan 23
Publication date: 2019
Volume: 7
Electronic Location ID: e6277
Received 2018 Mar 12; Accepted 2018 Dec 11
Copyright: ©2019 Murphy et al.
Copyright year: 2019
Copyright holder: Murphy et al.
License: This is an open access article distributed under the terms of the Creative Commons Attribution License, which permits unrestricted use, distribution, reproduction and adaptation in any medium and for any purpose provided that it is properly attributed. For attribution, the original author(s), title, publication source (PeerJ) and either DOI or URL of the article must be cited.
License URL: https://creativecommons.org/licenses/by/4.0/

Keywords: Flicker, Chick, Myopia, Refractive error, Refractive compensation, Sawtooth

Funding: The authors received no funding for this work.

==============================
Purpose

Myopia (short-sightedness) is the commonest visual disorder and greatest risk factor for sight threatening secondary pathologies. Myopia and hyperopia can be induced in animal models by rearing with optical lens defocus of opposite sign. The degree of refractive compensation to lens-induced defocus in chicks has been shown to be modified by directionally drifting sawtooth spatio-temporal luminance diamond plaids, with Fast-ON sawtooth spatio-temporal luminance profiles inhibiting the myopic shift in response to negative lenses, and Fast-OFF profiles inhibiting the hyperopic shift in response to positive lenses. What is unknown is whether similar sign-of-defocus dependent results produced by spatio-temporal modulation of sawtooth patterns could be achieved by rearing chicks under whole field low temporal frequency sawtooth luminance profiles at 1 or 4 Hz without a spatial component, or whether such stimuli would indiscriminately elicit a myopic shift such as that previously shown with symmetrical (or near-symmetrical) low frequency flicker across a range of species.

Methods

Hatchling chicks (n = 166) were reared from days five to nine under one of three defocus conditions (No Lens, +10D lens, or −10D lens) and five light conditions (No Flicker, 1 Hz Fast-ON/Slow-OFF sawtooth flicker, 4 Hz Fast-ON/Slow-OFF sawtooth flicker, 1 Hz Fast-OFF/Slow-ON sawtooth flicker, or 4Hz Fast-OFF/Slow-ON sawtooth flicker). The sawtooth flicker was produced by light emitting diodes (white LEDs, 1.2 –183 Lux), and had no measurable dark phase. Biometrics (refraction and ocular axial dimensions) were measured on day nine.

Results

Both 1 Hz and 4 Hz Fast-ON and Fast-OFF sawtooth flicker induced an increase in vitreous chamber depth that was greater in the presence of negative compared to positive lens defocus. Both sawtooth profiles at both temporal frequencies inhibited the hyperopic shift in response to +10D lenses, whilst full myopic compensation (or over-compensation) in response to −10D lenses was observed.

Conclusions

Whole field low temporal frequency Fast-ON and Fast-OFF sawtooth flicker induces a generalized myopic shift, similar to that previously shown for symmetrical sine-wave and square-wave flicker. Our findings highlight that temporal modulation of retinal ON/OFF pathways per se (without a spatial component) is insufficient to produce strong sign-of-defocus dependent effect.

Introduction

Most species are born with hyperopic eyes that, with normal visual experience, increase in size during early development to achieve emmetropia (i.e., no refractive error) (Norton & Siegwart Jr, 1995; Rabin, Van Sluyters & Malach, 1981; Stone et al., 1995; Troilo & Wallman, 1991; Wildsoet, 1997). This emmetropization process is controlled by complex gene-environment interactions (Fan et al., 2014; Morgan & Rose, 2005; Verhoeven et al., 2013). Epidemiological studies have demonstrated that exposure to environmental risk factors (such as spending limited time outdoors (Rose et al., 2008) or doing prolonged near-work (Huang, Chang & Wu, 2015)) can predispose the eye to excessive growth resulting in myopia (short-sightedness). Myopia is now the most common ocular disorder (Schneider et al., 2010) with rapidly increasing prevalence and severity worldwide (Lee et al., 2013; Seet et al., 2001), making the development of treatments to control ocular growth a health and socioeconomic priority.

Ocular growth has been extensively studied in animals, where myopia can be induced by rearing with negatively powered defocusing lenses affixed over the eye or form deprivation via lid suture or plastic occluders (Hodos & Kuenzel, 1984; Irving, Sivak & Callender, 1992; Schaeffel, Glasser & Howland, 1988; Wallman, Turkel & Trachtman, 1978; Wiesel & Raviola, 1977), and hyperopia (long-sightedness) can be induced by positive lenses (Irving, Sivak & Callender, 1992; Schaeffel, Glasser & Howland, 1988). Ocular growth and refractive shifts in these animal models can also be mediated by pharmacological manipulations of retinal ON/OFF pathway activity (Crewther & Crewther, 1990; Crewther, Crewther & Xie, 1996; Ehrlich et al., 1990; Smith, Fox & Duncan, 1991). Suppression of ON-pathway activity by L-α-aminoadipic acid in chicks and kittens decreases axial growth in normally developing animals (Crewther & Crewther, 1990; Crewther, Crewther & Xie, 1996; Crewther & Crewther, 2003). Conversely, OFF-pathway suppression by D- α-aminoadipic acid increases axial growth in normally developing chicks (Crewther & Crewther, 1990), and inhibits optically-induced growth changes (Crewther & Crewther, 1990; Crewther, Crewther & Xie, 1996; Crewther & Crewther, 2003). More recent gene knockout studies have demonstrated an increased susceptibility to FD in mice with ON-pathway defects (Chakraborty et al., 2015; Pardue et al., 2008), consistent with the effects of ON-pathway suppression in the chick FD model (Crewther & Crewther, 1990). Mice with an OFF-pathway defect did not develop myopia in response to FD (although this latter result may reflect the genetic background) (Chakraborty et al., 2014).

Although pharmacological and genetic studies have provided strong evidence implicating retinal ON/OFF pathways in ocular growth and refractive change, fewer studies have investigated the effects of visual stimuli designed to preferentially activate ON- or OFF-systems. One of the first studies was a notable paper by (Schwahn & Schaeffel, 1997) who raised chicks in flickering light of different temporal frequencies (12 and 6 Hz) and different duty cycles (4–75% ratio of light to dark periods) produced by rotating chopper disks. Not surprisingly, it was found that the extent to which FD and defocus-induced myopia could be suppressed was related to the length of the duty cycle, suggesting that myopia is related to dark duration and OFF pathway activation. Interestingly, there was no observable suppression of hyperopia.

In 2002, Crewther and Crewther (Crewther & Crewther, 2002) reared chicks in an environment lit with directionally moving sawtooth diamond plaids based on the human Brücke–Bartley Effect whereby a moving Fast-ON/Slow-OFF plaid has a perceptual darkening effect, and a Fast-OFF/Slow-ON spatiotemporal luminance profile creates a brightening effect (Cavanagh & Anstis, 2018; Cavanagh & Anstis, 1986). Chicks reared with Fast-ON sawtooth plaid illumination compensated fully to positive, but not to negative lenses. Conversely, chicks reared with Fast-OFF sawtooth plaid illumination compensated fully to negative, but not positive, lenses (Crewther & Crewther, 2002). More recently, Riddell et al. (2016) examined the electrophysiological basis of the Brücke-Bartley Effect in toad eye cups recordings and showed a sustained direct-current (DC) trans-tissue potential to drifting gratings. As in the human experience of moving sawtooth gratings, the sustained DC potential effect was greater for Fast-OFF (brightening effect) compared to Fast-ON sawtooth (darkening effect), suggesting that the asymmetries in DC potential originate in the photoreceptoral response to Fast-ON and Fast-OFF profiles. Thus, the ability of this paradigm to inhibit growth in a sign-of-defocus dependent manner that allows for potential localisation of structural origins, suggests that environmental modulation of the spatio-temporal activity of the ON/OFF pathways could represent a promising non-invasive treatment avenue for refractive errors using therapies aimed at the outer retina, as suggested by Crewther (2000).

What remains to be seen is whether the same inhibitory effects on refractive compensation produced by spatio-temporal modulation of sawtooth patterns, can be achieved using whole field, low temporal frequency sawtooth luminance profiles at 1 or 4 Hz without a spatial component, or whether such stimuli would indiscriminately elicit a myopic shift such as that previously shown to result from symmetrical (or near-symmetrical) low frequency flicker across a range of species (Rucker et al., 2015; Cheng et al., 2004; Cremieux et al., 1989; Crewther et al., 2006; Di et al., 2013a; Di et al., 2013b; Di et al., 2014; Li et al., 2016; Luo et al., 2017; Pan et al., 2017; Wang et al., 2013; Yu et al., 2011; Zhi et al., 2013).

In an attempt to resolve this question, Crewther et al. (2006) investigated the effect of whole field, low frequency flicker and found a myopic shift for all lens conditions with a stimulus comprised of an asymmetric light pulse similar to that of a Slow ON/Fast OFF sawtooth and 1:2 ratio light:dark cycle. However, due to the extended dark period in the stimulus profile created by the use of a halogen globe, the stimulus utilized in this study failed to replicate a true temporal illuminance-based Brücke–Bartley effect with a reversible sawtooth profile as produced in the 2002 study by the same lab, preventing an examination of the effect of continuous full field light dark transitions on directional growth following suppression of either the ON or OFF pathways.

Thus, the present study aimed to re-investigate the effects of whole field low frequency Fast-ON and Fast-OFF sawtooth flicker on refractive compensation to defocusing lenses in chick using the white light emitting diodes (LEDs). LEDS are a more precise system for controlling temporal luminance modulation and able to deliver reversible asymmetric light profiles (Fast-ON/Slow-OFF and Fast-OFF/Slow-ON) that closely mimicked the temporal aspects of the spatiotemporal sawtooth plaid paradigm (Crewther & Crewther, 2002).

Materials & Methods

Animals and rearing

A total of 166 male hatchling chicks (Leghorn/New Hampshire) were obtained from a local hatchery and raised from day five to nine in a light (12/12 h day/night cycle) and temperature (30 ± 0.5 °C) controlled enclosure (height: 0.5 m, length: 1.0 m, width: 0.75 m). On day five, chicks were fitted with either a monocular +10D or −10D lens, or left as No Lens controls. These defocusing goggles were made from modified human PMMA contact lenses (8.1 mm in diameter) attached to Velcro© and affixed to the periocular feathers of the right eye. Within each lens condition, chicks were randomly assigned to one of five light profiles: normal light (referred to herein as No Flicker [NF]), 1 Hz Fast-ON/Slow-OFF flicker, 4 Hz Fast-ON/Slow-OFF flicker, 1 Hz Fast-OFF/Slow-ON flicker, or 4 Hz Fast-OFF/Slow-ON flicker. All light profiles followed a 12/12 h day/night cycle.

The five light profiles were delivered using LEDs mounted to the roof of the enclosure. Ambient light was maintained constantly at 183 lux during the 12 h day cycle (No Flicker) or modulated (using the same lamp) at frequencies of either 1 Hz or 4 Hz where light levels varied between a minimum of 1.2 lux to a maximum of 183 lux, with an average light level of 97 lux. Ambient lighting in the No Flicker condition was maintained at the maximum (rather than the mean) of the flicker conditions to facilitate comparison with our previous study with a ‘no flicker’ condition of 180 lux (Crewther et al., 2006). For the flicker conditions, the LEDs produced a whole field sawtooth temporal luminance profile with a strict linear rise or fall in luminance achieved through feedback control circuitry connected to a DC power supply with computer input to implement waveform and frequency (for specifications please refer to Supplemental Information). These white LEDs work through a Blue LED driving a fluorescent material, ensuring that there is absolutely no change in the spectral distribution with brightness. Figure 1 shows the sawtooth wave profile recorded using a PowerLab (AD Instruments, Melbourne, Australia) via a light probe situated in the animal enclosure.

Figure 1 Light probe recordings of Fast ON/Slow OFF (Blue) and Fast OFF/Slow ON (Red) sawtooth flicker at a frequency of (A) 1 Hz, and (B) 4 Hz.

Biometric analysis

On day nine, chicks were anaesthetized (Ketamine 45 mg/kg: Xylazine 4.5 mg/kg i.m.) and both eyes were refracted by retinoscopy (Keeler, Vista Diagnostic Instruments) and axial dimensions were obtained from the average of at least three A-Scan ultrasonography traces (A-Scan III, TSL, 7 MHz probe; Teknar, Inc. St Louis, MO, USA). Each trace from the A-Scan provided peaks indicating the length of the eye (anterior pole of the cornea to inner limiting membrane of the retina), the distance from the posterior pole of the lens to the inner limiting membrane of the retina and the depth of the vitreous chamber (all measured in millimeters); subtracting the lens to retina measurement from the axial length gave a measure of the depth central to the axis of the anterior chamber. All procedures were conducted in strict accordance with La Trobe University Animal Ethics Committee guidelines (Approval Number 08/30) and adhered to the NHMRC of Australia and ARVO Statements for the use of Animals in Ophthalmic and Vision Research.

Data analysis

To control for differences in eye size within groups, refractive and biometric measures were converted to raw score differences between experimental and fellow eyes. To assess whether changes in ocular size were due to the experimental manipulation a three (lens condition × frequency × light profile) way ANOVA was utilized. Student Newman Keuls or Games-Howell post hoc tests were performed as appropriate.

Results

Table 1 reports the raw scores for refraction and axial length (mean and standard error of the mean) for each experimental group.

Table 1 Mean (±S.E) refraction and axial length of experimental eyes (EE) and fellow eyes (FE) for each lens, light profile and frequency condition.

		−10 D	No Lens	+10 D	
		n	EE RE (D)	FE RE (D)	EE AL (mm)	FE AL (mm)	n	EE RE (D)	FE RE (D)	EE AL (mm)	FE AL (mm)	n	EE RE (D)	FE RE (D)	EE AL (mm)	FE AL (mm)	
Fast OFF	1 Hz	10	−12.6 ±1.04	.30 ± .15	9.32 ± .07	8.81 ± .06	9	−.56 ± .47	−.14 ± .28	8.81 ± .06	8.75 ± .05	9	0.75 ±1.15	.17 ± .12	8.92 ± .12	8.85 ± .06	
	4 Hz	11	−7.18 ±1.68	.14 ± .30	9.03 ± .07	8.66 ± .05	9	−.28 ± .41	.11 ± .29	8.65 ± .06	8.60 ± .06	11	5.56 ± .39	.52 ± .24	8.43 ± .05	8.66 ± .04	
No Flicker		10	−8.4 ± .29	1.25 ± .21	9.20 ± .05	8.74 ± .05	19	.64 ± .25	1.01 ± .22	9.60 ± .04	9.55 ± .04	11	9.68 ± .29	1.30 ± .18	8.45 ± .07	8.75 ± .05	
Fast ON	1 Hz	11	−8.43 ± 1.04	.32 ± .23	9.23 ± .10	8.70 ± .08	11	.89 ± .33	.95 ± .34	8.76 ± .04	8.68 ± .04	12	5.62 ±1.17	.17 ± .31	8.80 ± .10	8.81 ± .06	
	4 Hz	11	−8.70 ±1.43	.10 ± .27	9.13 ± .10	8.73 ± .06	10	.60 ± .58	.62 ± .21	8.71 ± .07	8.65 ± .06	12	6.38 ±1.20	.75 ± .22	8.50 ± .04	8.70 ± .06	

Figure 2 illustrates the mean differences (Experimental –Fellow Eye) in biometric measures across groups.

Figure 2 Mean difference (±Standard Error S.E) in biometric measures between experimental and fellow eyes across lens and light conditions.

(A) Refractive difference (diopters), (B) axial length difference (mm), (C) vitreous chamber difference (mm), and (D) anterior chamber difference (mm). Asterisks indicate significant differences between condition for post hoc analysis.

As can be seen in Fig. 2A, all chicks wearing −10D lenses developed myopia under all light conditions. Hyperopic refractive compensation was also seen in eyes reared with +10D lenses under all flicker conditions. Fast OFF flicker at a frequency of 1 Hz produced the most notable impact on refraction, with eyes reared with −10D lenses under this condition displaying approximately 2 D more myopia than other −10 D lens groups. Fast OFF flicker also resulted in the greatest suppression of hyperopia in eyes wearing +10D lenses. Chicks reared without a lens were largely plano under all light conditions. Simple main effects analysis revealed a significant effect of light profile (F1,149 = 10.62, p = .0001), lens (F2,149 = 453.65, p < .0001) and frequency (F2,149 = 13.32, p < .0001). Significant interactions were also observed between lens and light (F2,149 = 3.89, p = .023), lens and frequency (F2,149 = 3.98, p = .021), light and frequency (F1,149 = 13.17, p = .0001), and lens and light and frequency (F2,149 = 3.28, p = .04).

Post hoc analysis comparing the effect of flicker frequency on refractive state revealed that eyes reared with −10D lenses were significantly more myopic under Fast OFF light conditions with 1 Hz flicker, in comparison to 4 Hz and no flicker (NF) conditions (see Table 2). No difference in refraction was found between flicker frequencies with a Fast ON sawtooth profile for eyes reared with −10D lenses. Under Fast OFF sawtooth conditions, chicks wearing +10D lenses were significantly less hyperopic when raised with 1 Hz flicker compared to 4 Hz and no flicker conditions (see Fig. 2A). Chicks reared under Fast OFF light were significantly less hyperopic than all other chicks wearing +10 D lenses. Refractions were significantly different between all lens groups when compared within frequency and sawtooth conditions. Comparisons between the effect of the sawtooth profile on refraction revealed that Fast OFF with 1 Hz flicker resulted in significantly greater myopia than Fast ON + 1 Hz or NF light conditions with −10D lenses, and significantly less hyperopia with +10D lens-wear. A summary of post hoc analyses is presented in Table S1.

Table 2 Summary of within light profile condition and within frequency condition post hoc interactions.

Sawtooth profile	Frequency	
Fast OFF/Slow ON	−10D Condition	1 Hz <(4 Hz No Diff. NF)	1 Hz	−10D Condition	Fast OFF <(Fast ON No Diff. NF)	
	No Lens Condition	1 Hz No Diff. NF No Diff. 4 Hz		No Lens Condition	Fast OFF No Diff. Fast ON. Fast OFF No Diff. NF	
	+10D Condition	1 Hz <4 Hz <NF		+10D Condition	Fast OFF <(Fast ON No Diff. NF)	
Fast ON/Slow OFF	−10D Condition	1 Hz No Diff. NF No Diff. 4 Hz	4 Hz	−10D Condition	Fast OFF No Diff. Fast ON. Fast OFF <NF	
	No Lens Condition	1 Hz No Diff. NF No Diff. 4 Hz		No Lens Condition	Fast OFF No Diff. Fast ON. Fast OFF No Diff. NF	
	+10D Condition	(1 Hz No Diff. 4 Hz) <NF		+10D Condition	(Fast OFF No Diff. Fast ON) <NF	
Notes.

< denotes more negative refraction (i.e., more myopic)

NF No Flicker

No Diff No significant difference between conditions

An overall increase in axial length in response to −10D lenses was observed, with eyes reared under Fast OFF flicker at 1 Hz showing the longest eyes (Fig. 2B). Fast OFF flicker at 1 Hz also resulted in longer axial length with +10D lens-wear in comparison to NF control, as did Fast ON flicker at 1 Hz. No Lens chicks showed slight increases in length for all conditions. As expected, all other +10D lens groups showed an overall decrease in axial length compared to No Lens groups. Simple main effects analysis revealed a significant effect of lens (F2,149 = 131.00, p < .0001) and frequency (F2,149 = 13.53, p < .0001). A significant interaction effect was obtained for lens and frequency (F2,149 = 4.46, p = .013). Post hoc analysis confirmed the observation that 1 Hz flicker for both sawtooth profiles led to longer axial lengths for +10D lens conditions.

Consistent with axial length measures, Fig. 2C shows an overall increase in vitreous chamber depth in response to −10D lenses for all light conditions, and a notable decrease in vitreous chamber depth in response to +10D lenses for all light conditions except for Fast OFF and Fast ON 1 Hz flicker in comparison to No Lens groups. No Lens chicks showed a slight increase in vitreous chamber depth. Simple main effects analysis revealed a significant effect of lens (F2,149 = 158.01, p < .0001) and frequency (F2,149 = 13.37, p < .0001), and a significant interaction between lens and frequency (F2,149 = 4.47, p = .013). Post hoc analysis again confirmed the observation that 1 Hz flicker for both sawtooth profiles led to longer vitreous chamber depths for +10D lens conditions.

As expected, a great degree of variability was observed in anterior chamber depth in response to lens wear (Fig. 2D). A significant interaction between lens and frequency (F2,149 = 5.88, p = .004) was found. Post hoc tests showed that 4 Hz flicker for both Fast OFF and Fast ON profiles led to shallower anterior chamber depths for +10D lens conditions.

Discussion

Previous studies have shown that low frequency sine-wave and square-wave flicker induces a generalized myopic shift in a range of species (Rucker et al., 2015; Cheng et al., 2004; Cremieux et al., 1989; Di et al., 2013a; Di et al., 2013b; Di et al., 2014; Li et al., 2016; Luo et al., 2017; Yu et al., 2011; Zhi et al., 2013). In the present study, we demonstrate that whole field temporal Fast ON and Fast OFF sawtooth flicker with no spatial component or dark phase induces a general increase in vitreous chamber depth and a subsequent myopic shift that is greater in the presence of defocus (particularly positive defocus), and greater at lower temporal frequencies (i.e., 1 Hz versus 4 Hz). The largest effects occurred for 1 Hz Fast OFF flicker, which strongly inhibited refractive compensation to positive lenses and induced over-compensation to negative lenses. Although this latter finding is in agreement with our previous study of spatiotemporal sawtooth flicker (Crewther & Crewther, 2002), on the whole, these results suggest that whole field temporal modulation of retinal ON/OFF pathways does not produce the strong sign-of-defocus dependent effects previously demonstrated for spatiotemporal ON/OFF stimuli (Crewther & Crewther, 2002).

As light levels can affect responses to lens and occluder wear (Norton & Siegwart Jr, 2013), further research is needed to rule out a contribution from differing mean illuminance between the temporal sawtooth in the present study and the spatiotemporal sawtooth in our previous study (97 and 387 lux, respectively), and between the no flicker and flicker conditions in the present study (183 and 97 lux, respectively). Past research has shown that chicks reared under very dim lighting (≤50 lux) for weeks to months develop longer axial lengths and a more myopic refraction (Bercovitz, Harrison & Leary, 1972; Cohen et al., 2011; Cohen et al., 2012; Lauber & Oishi, 1987). However, shorter duration studies have shown no difference in refractive development following 4 days of occluder wear under 50 lux versus 500 lux lighting (Ashby, Ohlendorf & Schaeffel, 2009), or following 7 days of wearing neutral density filters (0.5–1 log unit attenuation) versus clear filters under 500 lux lighting (Feldkaemper et al., 1999). These latter findings suggest that the contribution of mean illuminance to refractive outcomes across conditions is likely to be minimal. That said, a comparison of the present findings with those of our past study investigating near-symmetrical flicker (Crewther et al., 2006) is not inconsistent with a contribution from mean luminance in driving refractive change. This past study investigated the effects of flicker with a gradual rise and slightly sharper decline (approximating a Fast-OFF profile), and variable duration dark phase (longest for 1 Hz flicker and shortest for 4 Hz flicker) (Crewther et al., 2006). In this previous study, the 1 Hz flicker condition (which had the lowest mean luminance) caused the strongest inhibition of compensation to positive lenses, and all frequencies caused a similar degree of over-compensation to negative lenses. The refractive shifts observed in the present study, where there was no measurable dark phase and thus a slightly higher mean luminance, were similar but less extreme. The results of these two studies are also interesting in the context of previous research showing a correlation between the degree of myopia suppression by flicker and the duration of the dark phase at high frequencies (with longer dark phases conferring greater suppression) (Schwahn & Schaeffel, 1997). Our previous study (Crewther et al., 2006) demonstrated equivalent over-compensation to negative lenses despite large differences in the duration of the dark phase at 1, 2, and 4 Hz. These results demonstrate that the correlation between the flicker dark phase and the degree of myopia induction does not hold at low temporal frequencies when the mean illuminance is not controlled.

Previous physiological research in monkey (Khan et al., 2005) and toad eye cup (Riddell et al., 2016) indicates that temporal and spatiotemporal sawtooth modulation produces complex post-receptoral ON/OFF pathway asymmetries, with push-pull interactions evident between ON- and OFF- bipolar cell contributions to Fast ON and Fast OFF waveforms in both paradigms (Khan et al., 2005; Riddell et al., 2016). Although both paradigms mediate ON/OFF pathway activity in a polarity-specific manner, their effects on gross trans-retinal potentials are vastly different. Spatiotemporal sawtooth produces a sustained trans-retinal potential increase (presumably resulting from the summation of potentials from different local retinal regions), that has asymmetries at the photoreceptor-RPE level as well as post-receptorally (Riddell et al., 2016). The important difference between the Crewther & Crewther’s (2002) stimuli and stimuli utilised in the current study is the removal of the spatial aspects of each local diamond stimulus. The current stimulation procedure is a full field sawtooth ON to OFF, or OFF to ON temporal modulation of the environment. The neural response to whole field temporally modulated light is an ‘entire retina’ photoreceptor response that results in a global change in retinal transmembrane potential (Khan et al., 2005). In comparison, the neural response to the spatial plaid components of the diamond pattern is expected to allow relatively localized spatial and temporal buffering of the change of potential across the retina (Riddell et al., 2016). Such differences in their effects can be conceptualized in terms of spatial buffering of ions. Each change in illuminance, whether temporal only or spatiotemporal, results in a large efflux of sodium ions into the outer segments and potassium out in the subretinal space during darkness and reversal of this process during light onset (See Bialek & Miller, 1994; Gallemore, Hughes & Miller, 1997; Hamann, 2002; Li et al., 1994; Steinberg, 1985; Steinberg, Linsenmeier & Griff, 1985). It is proposed that the reduced effect 4 Hz flicker on eye growth is related to the refractory period required to modulate directional change in the buffering of ion distributions in conjunction with the magnitude of the change in transretinal potential caused by the flicker stimulus. Thus, the full field stimulation of the current 2018 study forces ionic changes in the radial dimension, reducing the processes to a 1-dimensional situation across the retina. In comparison, the 2002 spatiotemporal diamonds could allow buffering in the direction tangential to the retinal layers, for example, in the subretinal space, or indeed within individual RPE cells to produce a directional growth signal in response to signed defocus. Further investigation is warranted to determine the nature of the refractory period for ionic spatial buffering and the impact of spatiotemporal versus temporal frequency signals in driving signed compensation.

Conclusions

The present study demonstrated that both whole field low temporal frequency fast ON and fast OFF flicker induce a generalized myopic shift in chicks wearing defocusing lenses, but do not affect refractive development in chicks not wearing lenses. These results are highly similar to our previous findings using near-symmetrical flicker approximating a Fast OFF profile (Crewther et al., 2006), as well as being broadly concordant with the myopic shift shown following exposure to symmetrical flicker in chicks (Rucker et al., 2015), mice (Yu et al., 2011), guinea pigs (Cheng et al., 2004; Di et al., 2013a; Di et al., 2013b; Di et al., 2014; Li et al., 2016; Luo et al., 2017; Zhi et al., 2013), and cats (Cremieux et al., 1989). Most importantly, this study highlights the fact that temporal luminance modulation per se is not adequate to induce signed directional refractive compensation. Rather, signed modulation of refractive compensation requires a directional spatiotemporal component which likely effects relative adaptive brightness in local regions across the retina.

Supplemental Information

Supplemental Information 1 Raw biometric values

Click here for additional data file.

Supplemental Information 2 Specifications of LEDs used to generate sawtooth flicker profiles

Click here for additional data file.

The authors would like to thank Sarah Kiely for her assistance with animal rearing and initial analysis and manuscript development, and Loretta Giummarra for her assistance with animal rearing. This work was presented in part as a poster at the annual meeting for the Association for Research in Vision and Ophthalmology, ARVO, Fort Lauderdale, FL, USA, May 2010 (Kiely et al., 2010).

Additional Information and Declarations

Competing Interests

Author Contributions

Animal Ethics

Data Availability

The authors declare there are no competing interests.

Melanie J. Murphy conceived and designed the experiments, performed the experiments, analyzed the data, prepared figures and/or tables, authored or reviewed drafts of the paper, approved the final draft.

Nina Riddell prepared figures and/or tables, authored or reviewed drafts of the paper, approved the final draft.

David P. Crewther conceived and designed the experiments, prepared figures and/or tables, approved the final draft.

David Simpson designed and built stimulus generator, approved the final draft.

Sheila G. Crewther conceived and designed the experiments, performed the experiments, authored or reviewed drafts of the paper, approved the final draft.

The following information was supplied relating to ethical approvals (i.e., approving body and any reference numbers):

The protocols used in this study were approved by the La Trobe University Animal Ethics Committee (AEC 08/30).

The following information was supplied regarding data availability:

The raw data are available in a Supplemental File.

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
