# Peer review of "Temporal whole field sawtooth flicker without a spatial component elicits a myopic shift following optical defocus irrespective of waveform direction in chicks"

_PeerJ, doi:10.7717/peerj.6277_

## Round 0.1 · original submission · Major Revisions

Two reviewers have now reviewed the manuscript and provided feedback: one was primarily concerned about clarity of the language (and indeed supplied an annotated manuscript) and the other had a few minor concerns. Please address these issue line-by-line in your resubmission.

Reviewer 1 ·

Basic reporting

General
The senior authors published a paper in 2002 showing that saw-tooth shaped spatial/temporal luminance profiles, projected on the retina of chicks, have differential effects on the developmental compensation of the eyes of imposed positive or negative defocus. They attributed this selective stimulation of either the ON or OFF pathways in the retina. They found that predominant stimulation of ON channels had an inhibitory effect on eye growth while stimulation of OFF channels tended to increase it. A year later they published that pharmacological inactivation of ON or OFF channels had the opposite effects than what one might have expected from initial experiments: inactivation of the OFF channel increased eye growth while loss of ON input decreased eye growth. Obviously, light stimulation elicited not simply the reversed effects of pharmacological inactivation. While there was no simple explanation for this discrepancy, the authors later (2006) showed that low frequency (1-4 Hz) luminance modulation, with no specificity for ON or OFF, triggered increased eye growth in chicks, but only when they wore lenses.
In the current study, the authors refined their low frequency stimuli to more selectively stimulate ON or OFF channels. They find that, at 1 Hz, (1) both ON and OFF stimuli inhibited the compensation of positive lenses, (2) OFF stimulation increased myopia with negative lenses, (3) the effects were largely gone at 4 Hz. Apparently, full field On or OFF stimulation is different from local ON or OFF stimulation, as done in 2002. With full field stimulation, the major effect was relative myopia. While this is an interesting finding, I missed a discussion as to what the differences might for the retina between full field temporal ON or OFF stimulation, and the spatially distinct local ON or OFF stimulations. Also - would it be possible to explain why the 4 Hz stimulation had generally less effect than the 1 Hz stimulation, and how this might relate to the temporal frequency pattern on the retina in the study in 2002?
The authors refer to their “retina ion-drive efflux (RIDE)” model but it remained unclear to me how this might explain the different effects of full field ON-OFF versus mixed spatial-temporal ON-OFF stimulation. A paper by Riddell et al (on which they were senior authors) is cited but I believe that a short description why this paper can help to explain the findings would be helpful.
What surprises me is high variability in the data. They used 166 animals that were raised in 5 different light conditions which would be about 33 chicks per group (hope this correct, otherwise ignore). The SEMs on the axial length data were are in the range between 0.05 and 0.1 mm which would be a standard deviation of 0.05 to 0.1 *6 or 0.3 to 0.6 mm. This may be more than the total average change in axial length with lenses under normal lighting. It suggests that a some chicks have shown no effect and others the doubled effect. Basically, it look as if the different light stimulation paradigms may explain only a (small) part of the variance.

Specific
line 115. The illuminances were generally low, compared to regular office illuminances which should be in 500 lux. This may be a limitation of the LEDs? Or to better match the 2006 study? It is certainly possible that the effects of ON and OFF stimulation would be different under different illuminances. The authors point out that this should be studied in the future (line 301).
line 259ff. That extended sawtooth-shaped light exposure can damage the photoreceptors (even at 97 or 387 lux which is not very bright) is a unexpected assumption. Don’t photoreceptors see temporal modulation all the time?
line 120-125. The details about the voltages that are required for the LEDs seem unnecessary to me and could be omitted.
Figure 1. Could it be that Figure 1B shows the wrong profiles? They don’t look like “fast ON” or “fast OFF” but rather more like sine waves. If the profiles were like that it is not surprising at all that there were no differences between the effects of ON and OFF stimulation at 4 Hz.

Experimental design

The experiments with 1 Hz look alright to me but I was wondering why they considered the 4 Hz stimulus as selective for On or OFF

Validity of the findings

Low frequency low illuminance flicker causes myopia in chicks, with smaller impact of the exact wave form. May be a "chicken problem". Direct applicability to human myopia is not expected.

Sample sizes are larger than in most other experiments in chicks and the large variability of the effects is surprising

Stats is ok, and references complete

Discussion could also focus on a question listed in the comments to the authors.

Additional comments

see above

Reviewer 2 ·

Basic reporting

Intro and background are relatively clear. However, they can be bettered. Comments and questions about details are added to the original manuscript, figures, and figure legends.

The “Results” section is really hard to follow. I suggest that the authors only mention the more significant and relevant findings, and leave the details to figure legends, where readers can see the details while looking at the figure at the same time. Consider adding arrows pointing to the specific bars so readers can locate the more striking effect(s) immediately.

The “Discussion” is unenlightening. The authors did not identify the mechanisms by which whole-field, temporally intensity-modulated stimuli used here cause the observed results. Instead, they offered vague statements about “changes in ion and fluid flow”, “polarity effects in the outer retina, or “outer retinal asymmetries” – words and ideas that might explain anything … or nothing.

If there is indeed a proper basis of retinal circuit mechanisms and rational arguments, elaborate on them. Otherwise, focus on clearly documenting and describing the observed effects, while noting that it would be practically impossible to dissect retinal circuit functions in this way.

Regarding conclusions, it will be helpful if the authors tighten up the summary of relevant evidence, tighten the argument and the writing, and try to make one or two well-supported and reasonable conclusions.

Experimental design

As shown in Figure 1b, ON- and OFF sawtooth at 4Hz look similar, nor are they distinguishable from sine-wave waveform. How would this affect the results? This problem needs to be addressed.

Validity of the findings

no comment

Additional comments

no comment

Annotated reviews are not available for download in order to protect the identity of reviewers who chose to remain anonymous.

---

## Round 0.2 · Minor Revisions

The reviewers have a few more minor revision suggestions. Please address these completely.

Reviewer 1 ·

Basic reporting

General
The authors have raised chickens with -10 or +10D lenses under five different Ganzfeld illumination conditions, fast OFF at 1 and 4 Hz, no flicker, and fast ON at 1 and 4 Hz. Both fast OFF and fast ON generated a general shift into the myopic direction in all animals with lenses, but not in control chicks without lenses. They conclude that full field temporal modulation does not trigger directional effects on refraction and eye growth and that sign of defocus-specific effects require spatiotemporal components in the stimuli.
Most of my previous comments were appropirately addressed. However, I still have a number of more minor comments that would further improve the manuscript, as I believe.
Specific
line 48 - “small hyperopic eyes” in all species does not seem to reflect reality. Chick starts with 7 mm axial length which tree shrews reach much later, and see mice vs horse ...
line 53 - is Wallman 1978 a good reference for complex gene-environment interactions?
lines 63-65 - many of the cited papers here were not dealing with positive lenses but are nevertheless used as a reference for induction of hyperopia
line 83, typo “Schwahn”
line 106, just a comment: while I understand that the authors propose that temporal ON and OFF stimulation may have its effect on refractive error development “aimed at the outer retina”, one should not ignore that at least the ganglion cell receptive fields have also an antagonist spatial receptive field structure. If the authors propose “spatial-temporal” activity of the ON/OFF pathways to inhibit refractive errors, this should also involve the inner retina
line 122ff, it is good that the temporal brightness profiles of the LEDs were measured but It would be good to know why the slow rising or decaying phases have the small steps - is this a issue of the measurement device, or does the light sources really emit at these steps?
line 132ff regarding the experiments in 2002 - could the assymmetrical OKN in left and right eyes and also the head optomotor responses have an effect on the temporal luminance profiles that end up on the retina? It is possible that non-stationary stimuli (like in the drum in 2002) may stimulate retinal detectors differently from stationary stimuli with defined temporal luminance profiles.
line 164. It is not very common to refract chicks under anesthesia. What drug was used and is it possible that it changed refractive states/accommodation tonus?
line 194ff. The description is not very straightforward. Perhaps better: “As can be seen in Figure 2, all chicks wearing -10D lenses developed myopia under all light regimens. Treatment with +10D lenses resulted in hyperopia in all chicks, except for the 1 Hz OFF stimuli.”
line 250. Britton et al 2013, 2014 are ARVO abstracts - should this be mentioned? In line 334, it is said that some of the the current study was presented at ARVO in 2010 but a reference is not provided.
line 267. The illuminance topic is important and it would be interesting to compare the illuminances (or luminances when it was a screen for stimulation) in related studies.
line 308 ff. It would be better to limit speculations. Why is the refractory period in the range of 250 msec (4Hz) - was this measured?
line 318 perhaps better “the present study demonstrated that both whole field low temporal frequency fast ON and fast OFF flicker ...” and, perhaps better: “... but does not affect refractive development in chicks not wearing any lenses”
line 323ff - it would be better to mention the animal model in each case for these references
references. Crewther et al 2005 is cited twice

Experimental design

some minor questions were left - see my comments above

Validity of the findings

data seem properly collected and the statistical noise is realistic

Additional comments

General
The authors have raised chickens with -10 or +10D lenses under five different Ganzfeld illumination conditions, fast OFF at 1 and 4 Hz, no flicker, and fast ON at 1 and 4 Hz. Both fast OFF and fast ON generated a general shift into the myopic direction in all animals with lenses, but not in control chicks without lenses. They conclude that full field temporal modulation does not trigger directional effects on refraction and eye growth and that sign of defocus-specific effects require spatiotemporal components in the stimuli.
Most of my previous comments were appropirately addressed. However, I still have a number of more minor comments that would further improve the manuscript, as I believe.
Specific
line 48 - “small hyperopic eyes” in all species does not seem to reflect reality. Chick starts with 7 mm axial length which tree shrews reach much later, and see mice vs horse ...
line 53 - is Wallman 1978 a good reference for complex gene-environment interactions?
lines 63-65 - many of the cited papers here were not dealing with positive lenses but are nevertheless used as a reference for induction of hyperopia
line 83, typo “Schwahn”
line 106, just a comment: while I understand that the authors propose that temporal ON and OFF stimulation may have its effect on refractive error development “aimed at the outer retina”, one should not ignore that at least the ganglion cell receptive fields have also an antagonist spatial receptive field structure. If the authors propose “spatial-temporal” activity of the ON/OFF pathways to inhibit refractive errors, this should also involve the inner retina
line 122ff, it is good that the temporal brightness profiles of the LEDs were measured but It would be good to know why the slow rising or decaying phases have the small steps - is this a issue of the measurement device, or does the light sources really emit at these steps?
line 132ff regarding the experiments in 2002 - could the assymmetrical OKN in left and right eyes and also the head optomotor responses have an effect on the temporal luminance profiles that end up on the retina? It is possible that non-stationary stimuli (like in the drum in 2002) may stimulate retinal detectors differently from stationary stimuli with defined temporal luminance profiles.
line 164. It is not very common to refract chicks under anesthesia. What drug was used and is it possible that it changed refractive states/accommodation tonus?
line 194ff. The description is not very straightforward. Perhaps better: “As can be seen in Figure 2, all chicks wearing -10D lenses developed myopia under all light regimens. Treatment with +10D lenses resulted in hyperopia in all chicks, except for the 1 Hz OFF stimuli.”
line 250. Britton et al 2013, 2014 are ARVO abstracts - should this be mentioned? In line 334, it is said that some of the the current study was presented at ARVO in 2010 but a reference is not provided.
line 267. The illuminance topic is important and it would be interesting to compare the illuminances (or luminances when it was a screen for stimulation) in related studies.
line 308 ff. It would be better to limit speculations. Why is the refractory period in the range of 250 msec (4Hz) - was this measured?
line 318 perhaps better “the present study demonstrated that both whole field low temporal frequency fast ON and fast OFF flicker ...” and, perhaps better: “... but does not affect refractive development in chicks not wearing any lenses”
line 323ff - it would be better to mention the animal model in each case for these references
references. Crewther et al 2005 is cited twice

·

Basic reporting

The manuscript is clear and well written

Experimental design

Design is correct, with a small question: the non-flicker and flickering conditions were carried out with two differences mean luminances (183 vs 97 lux), with the max luminance of the flicker conditions matching the mean luminance of the non-flicker. Why was the maximum matched and not the mean value? The discussion of the article cites potential effects of different mean luminances.

Validity of the findings

Data analysis is sound. Conclussions are clear and well-supported.

---

## Round 0.3 · accepted · Accept

Thank you for addressing the issues raised by the authors. I will recommend publication to the board with no further revisions.

#